# UV-C Treatment Maintains the Sensory Quality, Antioxidant Activity and Flavor of Pepino Fruit during Postharvest Storage

**DOI:** 10.3390/foods10122964

**Published:** 2021-12-02

**Authors:** Yaqi Zhao, Jinhua Zuo, Shuzhi Yuan, Wenlin Shi, Junyan Shi, Bihong Feng, Qing Wang

**Affiliations:** 1College of Agricultural, Guangxi University, Nanning 530004, China; zhaoyaqii@163.com (Y.Z.); shiwenlinn@163.com (W.S.); 2Key Laboratory of the Vegetable Postharvest Treatment of Ministry of Agriculture, Beijing Key Laboratory of Fruits and Vegetable Storage and Processing, Institute of Agri-Food Processing and Nutrition (IAPN), Beijing Academy of Agriculture and Forestry Sciences, Beijing 100097, China; zuojinhua@126.com (J.Z.); yuanshuzhi@nercv.org (S.Y.); shijunyan0130@126.com (J.S.); 3Key Laboratory of Biology and Genetic Improvement of Horticultural Crops (North China) of Ministry of Agriculture, Key Laboratory of Urban Agriculture (North) of Ministry of Agriculture, National Engineering Research Center for Vegetables, Beijing Academy of Agriculture and Forestry Sciences, Beijing 100097, China

**Keywords:** UV-C, pepino fruit, sensory quality, antioxidant enzymes, flavor

## Abstract

This study examines ultraviolet-C (UV-C) treatment supplementation as a means of inhibiting the senescence of pepino fruit after harvest. Pepino fruits were subjected to 1.5 kJ/m^2^ UV-C treatments and then packed and stored at 10 °C for 28 d. Results showed that 1.5 kJ/m^2^ UV-C treatment had the greatest ability to maintain firmness, and reduced the level of respiration and ethylene production. Further analysis indicated that the 1.5 kJ/m^2^ UV-C treatment maintained the content of total soluble solids (TSS), chlorophyll, vitamin C, flavonoids, and total phenolics. Lower levels of malondialdehyde (MDA) and higher levels of antioxidant enzyme activity were found in UV-C treated fruit during storage. An electronic nose (E-nose) and headspace-gas chromatography-mass spectrometry (HS-GC-MS) was used to determine volatile compounds. Results revealed that the UV-C treatment may promote the synthesis of a large number of alcohols and esters by maintaining the overall level of acids, aldehydes, and esters in fruits. This may contribute to the maintenance of the flavor of harvested fruits. In conclusion, 1.5 kJ/m^2^ UV-C treatment was demonstrated to be an effective treatment for the maintenance of the sensory, nutritional, and flavor parameters of pepino fruit.

## 1. Introduction

Pepino (*Solanum muricatum* Aiton) fruit is a solanaceous, vegetatively propagated fruit crop of Andean origin that has received increasing interest as an exotic fruit commodity [1,2]. Several countries have explored developing pepino into a new, horticultural product for fresh markets, and outlined the breeding objectives needed to adapt pepino production to new agroclimatic conditions and provide useful, fruit organoleptic traits [3]. Pepino fruits exhibit considerable variation in size, shape, and color depending on the cultivar. Some selections are very aromatic and juicy and are mainly used in dessert dishes or salads, whereas other selections are used in juices or milk drinks, similar to other exotic fruits [4,5]. Pepino fruits are low in calories, very rich in minerals, including calcium, phosphorus, and potassium, and rich in vitamins, such as thiamin, niacin, riboflavin, and vitamin C [5,6]. Medicinal uses, such as the treatment of hypertension, use as a diuretic, and as an antitumor treatment, have also been attributed to pepino fruit, further strengthening its commercial value [4,7,8]. One of the main problems with pepino fruit, however, is the significant decrease in organoleptic and nutritional quality that occurs due to poor handling and inadequate storage conditions, which greatly reduces the interest in this crop by commercial distributors [9,10]. In this regard, previous studies investigating storage and packaging technologies, such as film packaging [11], controlled atmosphere (CA) [9], and modified atmosphere (MA) [11], have found that specific gas concentrations can improve the storability and quality of pepino fruit.

Ultraviolet-C (UV-C) is considered a safe, effective, and economical treatment that can be used to maintain the postharvest quality of fruit and vegetables during storage [12,13]. Low doses of UV-C, ranging from 0.01 kJ/m^2^ to 39 kJ/m^2^, have been reported to elicit a hormetic response in a variety of fruit and vegetables [14,15]. Low-dose UV-C treatments have been reported to delay senescence, induce resistance to pathogens, maintain fruit flavor, and prolong the length of time produce can be stored with optimal quality [14,16,17]. UV-C irradiation has been used to treat a variety of fruit and vegetables, including strawberry, blueberry, grape, papaya, apple, carrot, tomato, broccoli, spinach, and others [18,19,20]. Despite the general interest in this technology, however, studies on the effect of UV-C irradiation on the postharvest quality of pepino fruit are lacking.

Previous research demonstrated that a 1 kJ/m^2^ UV-C treatment could effectively reduce chilling injury and maintain the flavor quality of fruits at 4 °C [21]. Based on the collective rsearch on UV-C, the objective of the present study was to determine the effect of UV-C irradiation on the postharvest quality of pepino fruit stored at 10 °C. Treatments comprising different doses of UV-C were compared to determine their effect on several quality parameters, including sensory evaluation, nutrient content, antioxidant enzyme activity, and volatile production. Results of our study provide a theoretical basis for the application of UV-C on pepino fruit and contribute to the further development of UV-C technology for the preservation of fruit and vegetables.

## 2. Materials and Methods

### 2.1. Plant Material and Treatments

Pepino fruit (*Solanum muricatum* Aiton, ‘Chang-li’) were harvested at commercial maturity from an agricultural field in Minqin County, Wuwei City, Gansu Province, China (longitude 103.08; latitude 38.62). The maturity standard used for harvesting was purple stripes on the fruit surface, and a TSS content of 6–7%. Each fruit was covered with a low-density polyethylene packaging net to avoid bruising, and then immediately packed in boxes and transported back to the laboratory. Fruit was selected for uniformity in size, color, firmness, shape, and the absence of evidence of any mechanical damage.

Pepino fruits removed from packaging were exposed to the irradiation of a UV-C lamp tube (30 W/T8, 254 nm, Philips, Amsterdam, Holland) at a distance of 25 cm. The UV-C dose was calculated based on irradiance, which was measured by a light-meter (UV DATAL, 115/230V, Cole-PARMER, Vernon Hills, IL, USA). Prior to conducting the first experiment, nine pepino fruit were randomly selected from amongst the harvested fruit and used as the control group (0 d). An additional 540 randomly selected fruit were randomly distributed into five groups, with approximately 108 fruits in each group and stored for 28 d. The treatments used on the five groups were as follows: (a) untreated fruit, serving as a control group; (b) 1.0 kJ/m^2^ dose of UV-C; (c) 1.5 kJ/m^2^ dose of UV-C; (d) 2.0 kJ/m^2^ dose of UV-C; (e) and 3.0 kJ/m^2^ dose of UV-C. After treatment, the fruit were enclosed in 0.03 mm thickness polyethylene film bags (every bag containing 36 fruit) and then stored in darkness at (10 ± 1) °C and 80–90% relative humidity. The fruits were collected every 7 days to determine various indexes. Each group was further divided into three biological replicates with three fruits for each replicate (*n* = 3). The pulp and peel of each fruit from were frozen separately in liquid nitrogen and stored at −80 °C. The pulp was used to determine peroxidase (POD) activity, ascorbate peroxidase (APX) activity, catalase (CAT) activity, vitamin C content, total phenolics, and flavor-related volatiles. The peel tissue was used to assess chlorophyll, flavonoid, anthocyanin and MDA content.

### 2.2. Sensory Scores

Evaluation of pepino fruit was conducted following a modification of the approach described by Pluda et al. [22]. Three fruits for each replicate were cut into four pieces of the same size and marked. A ten-member trained panel ranked the pepino fruit from the different treatment groups based on overall flavor and texture. The sensory quality of pepino fruit was scored on a scale from 1 to 5, where 5 was excellent (fresh, aromatic flesh without odor, full juice, high commercial value), 4 was good (dark color, no aroma and no peculiar smell of flesh, full juice, high quality, and commercial value), 3 was moderate (dark color, little flesh juice, soft, no odor, average quality and commercial value), 2 was poor (brown dark colored fruit, little flesh juice, very soft, peculiar smell, poor quality and no commercial value), and 1 was extremely poor (rotten, moldy, smelly, inedible, poor quality, no commercial value).

### 2.3. Firmness, Respiration Rate, and Ethylene Production

A GY-40J type digital fruit hardness tester (Zhejiang Top Instrument Co., Ltd., Hangzhou, China) was used to assess firmness. Three pepino fruits for each replicate were measured for firmness at the equator of each fruit. The diameter of the probe was 0.7 cm, and the average value of three readings from the equator of each fruit was recorded. The results of firmness were reported as Newton (N).

Three pepino fruits for each replicate were measured respiration rate and ethylene production. Respiration rate was assessed as the production of carbon dioxide (CO_2_). Each replicate pepino fruit were placed in a glass sealed jar for 1 h and respiration was then determined using a GHX-3051 respiration analyzer (Jun Fang Li Hua Technology-Research Institute of Beijing, Beijing, China). The results of respiration rate were reported as mg·kg^−^^1^·h^−^^1^.

Ethylene production was determined using a gas chromatograph (7820 A, Agilent Technologies, Inc., Santa Clara, CA, USA). Each replicate pepino fruit were placed in a sealed container with a gas extraction port. Headspace gas was extracted from a sample after 1 h and then 1 mL of gas was injected into a gas chromatograph. The level of ethylene was calculated based on the linear relationship between the peak area and the concentration of ethylene. Ethylene production is expressed as µmol·kg^−1^·h^−1^.

### 2.4. TSS, Chlorophyll, Vitamin C, Flavonoids, Anthocyanin, and Total Phenolics Content

3 pepino fruit for each replicate were measured the level of TSS, which was determined using a hand-held refractometer (Atago PAL-1, Tokyo, Japan) and reported as Brix (^O^%). Chlorophyll was extracted from frozen samples of pepino fruit peel tissues using a solution of 80% aqueous acetone and chlorophyll levels were measured as described by Sun et al. [23]. Briefly, the extract was centrifuged at 13,000× *g* for 10 min at 4 °C, and the absorbance of the supernatant was recorded at 645 nm and 663 nm using a UV-spectrophotometer (UV-1800, Shimadzu Corporation, Tokyo, Japan) and chlorophyll content were expressed as mg/g.

Vitamin C content was determined using an ammonium molybdate colorimetry assay as described by [24]. Absorbance at 760 nm was measured in a spectrophotometer and ascorbic acid was used to generate a standard curve. Vitamin C content was expressed as mg/g.

The flavonoid content was carried out in accordance with the previous study by Pirie et al. [25] with some modifications. A total of 2 g of pepino peel tissue and 5 mL methanol (0.1% hydrochloric acid) were ultrasonically extracted at 25 °C for 10 min, and then centrifuged at 13,000× *g* for 20 min at 4 °C, and methanol was added to bring it to a volume of 25 mL. The supernatant was used to evaluate the content of total phenols, flavonoids, and anthocyanins. A total of 1 mL supernatant and 0.5 mL of NaNO_2_ (1.5%) solution were mixed, then 1 mL 10% AlCl_3_ and 1 mL 1 M NaOH were added. After incubating for 5 min in the dark, the absorbance was measured at 490 nm. The results are expressed in mg of epicatechin equivalent per kg of fruit weight (mg/kg).

Total anthocyanin content was evaluated using the method of Zhang et al. [26]. Briefly, 0.5 mL supernatant and 2 mL of two different buffers (0.025 M KCl pH 1 and 0.4 M CH_3_COONa pH 4.5) were combined, then incubated in the dark for 15 min. Absorbance was then measured at 510 and 700 nm. Results are expressed in mg cyanide-3-O-glucoside equivalent per kg fruit weight (mg/kg).

Total phenol content was determined using the method of Esua et al. [27] with slight modification. Briefly, 0.04 mL supernatant and 1 mL of Folin–Ciocalteu reagent were mixed together, and a 7% Na_2_CO_3_ solution was then added to the mixture. After incubating for 1 h in the dark, absorbance was measured at 760 nm. Results are expressed as mg gallic acid equivalents per one kg fruit weight (mg/kg).

### 2.5. POD, APX, and CAT Activity, and MDA Content

POD, APX, and CAT activity, as well as MDA content, were measured using the method described by Cao et al. [28]. Frozen samples of powdered pepino pulp (1 g) were mixed with 5 mL phosphate buffer (0.05 M, pH 7.0), containing 0.001 M EDTA and 2% PVP, and centrifuged at 13,000× *g* for 30 min at 4 °C. The supernatants were used to determine POD and CAT activity. The reaction mixture for POD activity consisted of 0.1 mL supernatant, 1 mL phosphate buffer (pH 7.8), and 0.9 mL 2% guaiacol. The change in absorbance of the reaction mixture at 470 nm in 1 min was recorded. The reaction mixture for CAT activity was 1 mL 0.3% H_2_O_2_, 1.9 mL 0.1 M phosphate buffer (pH 7.8), and 0.1 mL supernatant. POD and CAT activities was measured along with the change of per gram tissue in absorbance at 470 and 240 nm over one minute, and were expressed as U.

The following procedure was used to measure APX activity. Samples of powdered pepino fruit pulp (1 g) were mixed with 5 mL of phosphate buffer (pH 7.5), containing 0.001 M EDTA, 0.001 M ascorbic acid, and 2% PVP. The solution was then centrifuged at 13,000× *g* for 30 min at 4 °C, and the resulting supernatant was used to determine APX activity. The reaction system for measuring APX activity was 2.6 mL of 0.05 M phosphate buffer (pH 7.5), 0.1 mL of supernatant, and 0.3 mL of 0.002 M H_2_O_2_. APX activity was measured, along with the change of per gram tissue in absorbance at 290 nm over one minute, expressed as U.

The following protocol was used to measure MDA content. Samples of powdered, frozen pepino peel (1 g) were mixed with 5 mL of trichloroacetic acid and centrifuged at 13,000× *g* for 20 min at 4 °C, and the supernatants were collected. Subsequently, 2 mL of trichloroacetic acid was added to each sample tube and placed in a boiling water bath for 20 min. The concentration of MDA in the supernatant was determined after cooling to room temperature by measuring absorbance at 450 nm, 532 nm, and 600 nm. MDA content was calculated using the following formula (1):MDA (M) = [6.45 × (OD_532_ − OD_600_) − 0.56 × OD_450_] × 10^−6^(1)

The content of MDA in each kilogram of fruit sample (fresh weight) was then calculated. Results are expressed in mol/g.

### 2.6. Electronic Nose (E-Nose) Analysis of Volatile Signatures

An E-nose (Airsense Analytics, GmBH, Schwerin, Germany) was used to evaluate emitted volatile signatures using the headspace inhalation method [29]. Three pepino fruits were placed in an airtight glass container to determine the E-nose footprint at each of the sampled time points. The containers, containing pepino fruit samples, were equilibrated for 60 min at 10 °C. Subsequently, an E-nose sample needle was inserted into the glass container to assess the volatile signature. Each measurement time was 120 s and repeated three times. The flow rate in the E-nose chamber during sampling was 300 mL·min^−1^ and the E-nose sensors were recalibrated with a cleaning gas for 80 s between each measurement. The E-nose was equipped with ten sensors. The sensors were W1C, W5S, W3C, W6S, W5C, W1S, W1W, W2S, W2W, and W3S, and were constructed to measure aromatic compounds, oxynitride, ammonia and aromatic compounds, hydrogen, alkane and aromatic compounds, methane, sulfur compounds, ethanol, aromatic and organic sulfur compounds, and alkanes, respectively.

### 2.7. HS-GC-MS Analysis of Volatiles

Flavor volatiles were measured according to the method from Aubert et al. [30] with modifications. Exactly 5 g pulp of each sample was placed in a 20 mL sealed vial with a PTFE-silicone septum (Supelco, Bellefonte, PA, USA) with 1.5 g NaCl, and the vials were then placed in an automatic sampling instrument after 1 h at room temperature. The analysis of the products was carried out by HS-GC-MS (GC–MSQP 2010 Plus system with Headspace Sampler 10; Shimadzu, Kyoto, Japan). Separation was performed on a polar (30 m × 0.25 mm × 0.25 µm) fused silica capillary column. Helium was used as the carrier gas at a flow rate of 1.5 mL·min^−1^. One microliter aliquot sample were injected at 250 °C in 46 split ratios. The temperature program for GC followed the sequence: hold for 4 min at 40 °C, increase to 50 °C at a speed of 5 °C·min^−1^, hold for 3 min, increase to 220 °C at a speed of 10 °C·min^−1^, hold for 2 min. The transfer line and ion source temperature were maintained at 150 °C and 200 °C, respectively. Qualitative analysis was performed in the electron impact (EI) mode using the full scan mode in the m·z^−1^ range of 45–550 amu. The unknown peaks were identified using the library of NIST 14 and NIST 14s. The external standard method was used to quantify the main components in pepino fruit. The GC conditions and MS conditions were the same as those described in the above GC-MS analysis. Using 3-Octanol in a concentration range of 0.1 μg/L~10 mg/L, a standard curve of the correlation between peak area and content was established (R^2^ > 0.99). The standard curve method was used to calculate the content of each of the identified compounds.

### 2.8. Statistical Analysis

Statistical analyses were conducted using SPSS 22 (SPSS Inc., Chicago, IL, USA). The E-nose data was analyzed using linear discriminant analysis (LDA). Additional data analysis and processing software utilization included Origin 2019 and Winmuster (version 1.6.2), provided along with the E-nose instrument. The data was subjected to a one-way ANOVA, and Duncan test was used to compare means between treatments. The level of significance was set at *p* < 0.05. Presented data represent the mean ± standard deviation. The heat map was generated in Microsoft Excel 2019 and PowerPoint 2019.

## 3. Results

### 3.1. Sensory Score, Firmness, Respiration Rate, and Ethylene Production

The sensory score rating of pepino fruit exhibited a steady decline over the course of storage (Figure 1A). The sensory score of pepino fruit that received the 1.5 kJ/m^2^ UV-C treatment was higher than the other treatment groups after 14 d of storage. At the 28th day of storage, the fruit in the control group had poor quality and no commercial value (score < 2), whereas the fruit in the 1.5 kJ/m^2^ UV-C group still had average quality and a commercial value (score > 3). The overall results indicated that the 1.5 kJ/m^2^ treatment had the greatest ability to maintain the sensory quality of pepino fruit during storage.

The firmness of pepino fruit also exhibited a downward trend during storage (Figure 1B). Firmness in the 1.0 kJ/m^2^ and 1.5 kJ/m^2^ UV-C treatment groups decreased gradually, whereas the firmness in the control and other treatment groups decreased sharply beginning on the day 21 of storage. Notably, the firmness of pepino fruit in the 1.5 kJ/m^2^ UV-C treatment group was higher than it was in the other treatment groups from the 14–28 d of storage. The overall data indicated that the 1.5 kJ/m^2^ treatment had the most positive effect on maintaining the firmness of pepino fruit during storage.

Respiratory rate initially decreased and then increased during storage (Figure 1C). The respiratory rate in the 1.5 kJ/m^2^ UV-C treatment group was lower than in the other treatment groups beginning at day 21 of storage. The respiration rate in the 1.5 kJ/m^2^ UV-C treatment group on day 28 of storage was 0.0155 mg·kg^−^^1^·h^−^^1^, whereas in the control and other treatment groups it was higher than 0.0215 mg·kg^−^^1^·h^−^^1^. Thus, the data indicate that the 1.5 kJ/m^2^ UV-C treatment had the greatest ability to inhibit the increase in respiration rate that occurred during the storage of pepino fruit.

The rate of ethylene production decreased during the first 14 d of storage and then remained relatively stable (Figure 1D). Ethylene production in the control group was higher than it was in the other treatment groups beginning at 7 d of storage. Ethylene production in the 1.5 kJ/m^2^ UV-C treatment was lower than it was in the other treatment groups on days 7 and 21 of storage, but no significant difference was observed between any of the other UV-C treatment groups. Based on the collective results, the 1.5 kJ/m^2^ UV-C treatment was selected for more detailed studies for its effect on the metabolism of pepino fruit.

### 3.2. The Level of TSS, Chlorophyll, Vitamin C, Flavonoids, Anthocyanin, and Total Phenolics

TSS content (Figure 2A) and chlorophyll content (Figure 2B) in pepino fruit of the control group increased during storage at 14 d but then decreased rapidly. TSS content in the control groups was only 5.60% on 28 d of storage and chlorophyll content was only 1.59 mg/g. TSS content in the 1.5 kJ/m^2^ UV-C treatment group also decreased during the first 14 d of storage, and then remained relatively stable. TSS content in the 1.5 kJ/m^2^ UV-C treatment group was 6.02% on day 28 of storage, which was higher than it was in the control group. Chlorophyll content in the 1.5 kJ/m^2^ UV-C group increased on day 7 of storage, and then exhibited a gradual downward trend during the remaining period of storage. Chlorophyll content was 1.89 mg/g in the 1.5 kJ/m^2^ UV-C group on day 28 of storage, which was 0.3 mg/g higher than in the control group.

The level of vitamin C in the 1.5 kJ/m^2^ UV-C treatment group, however, was higher than it was in the control group from days 7 to 28 of storage. The level of vitamin C decreased by 22% in the 1.5 kJ/m^2^ UV-C treatment group over the 28 d of storage, whereas it decreased by 27% in the control group (Figure 2C).

Changes in the levels of flavonoid and anthocyanin in the control and 1.5 kJ/m^2^ UV-C treatment groups are presented inFigure 2D,E. The content of flavonoids and anthocyanins in the control group increased and then decreased over the first 14 days of storage. Flavonoid and anthocyanin levels in the UV-C treatment group, however, were significantly higher than those in the control group after 28 days of storage.

The level of total phenolics in pepino fruit of the two treatment groups is shown in Figure 2F. Total phenolics exhibited a general decreasing trend in both groups overall, however, the level of phenolics in the 1.5 kJ/m^2^ UV-C treatment group was higher than it was in the control group on the day 14, 21, and 28. The content of total phenolics in the 1.5 kJ/m^2^ UV-C treatment group had decreased by 21% at the end of 28 d storage, whereas it decreased by 34% in the control group.

### 3.3. POD, APX and CAT Activity, and MDA Levels

POD (Figure 3A) and CAT (Figure 3C) activity in pepino fruit during storage in the control group and 1.5 kJ/m^2^ UV-C treatment group tended to increase overall. Notably, POD and CAT activity in the 1.5 kJ/m^2^ UV-C treatment group was always higher than it was in the control group. A difference between the control and UV-C treatment group in POD and CAT activity was evident from day 7 to 28. These results clearly indicate that the 1.5 kJ/m^2^ UV-C treatment induced a greater level of POD and CAT activity than was present in the untreated, control pepino fruit.

APX activity in the two groups exhibited an overall downward trend during storage (Figure 3B). APX activity in fruit that received the 1.5 kJ/m^2^ UV-C treatment, however, was higher than in the control group on day 7 and 28 of the storage. At the end of storage period, APX activity in the 1.5 kJ/m^2^ UV-C treatment group was 70% higher than it was in the control group.

In contrast to APX activity, MDA content exhibited an overall upward trend during storage in both groups (Figure 3D). Notably, MDA content in the pepino fruit of the 1.5 kJ/m^2^ UV-C treatment group increased sharply after day 21 of storage, whereas the control group exhibited increase on the day 14. MDA content in the control group at the end of storage was 0.072 times greater than it was in the 1.5 kJ/m^2^ UV-C treatment group.

### 3.4. Flavor-Related Parameters

#### 3.4.1. E-Nose Analysis Results

The flavor characteristics of pepino fruit were comprehensively analyzed using an E-nose equipped with 10 kinds of sensors. The results obtained with the E-nose are based on the odor molecules being emitted from the pepino fruit and their concentration. Representative E-nose sensor intensity curves of the volatile compounds are presented in Figure 4A,B. The differences in the profiles observed between the control group and 1.5 kJ/m^2^ UV-C treatment group suggest that a noticeable alteration in the composition of volatile compounds occurred in response to the UV-C treatment by the end of the storage period.

Linear discriminant analysis (LDA) was used to linearly transform the original data vector, so that samples with different properties could be more readily distinguished [31]. Two variables (LD1 = 78.309%, LD2 = 16.088%) are based on the E-nose values and collective values greater than 90% indicate that each group can be readily distinguished. The distance between the two groups represents the difference between the two groups. Results indicate that the extension of storage time was the main contributor to the observed changes in the profile of volatile compounds in pepino fruit, followed by UV-C treatment. Notably, the 1.5 kJ/m^2^ UV-C treatment group was closer to the initial volatile (odor) signature (0 d) than the control group.

A radar fingerprint chart of the volatile compounds in the initial control group (0 d), 1.5 kJ/m^2^ UV-C group, and the control group during storage is presented in Figure 4D. The closer the distance of a reading to its initial value, the less change that has occurred. Results indicate that the response values for the W5S, W1W, W2S, and W2W sensors exhibited a significant change, indicating that ethanol, oxynitride, and sulfur compounds in pepino fruit were altered in composition and/or quantity during storage.

#### 3.4.2. GC-MS Analysis Results

Volatile compounds in pepino fruit and the changes that occurred during storage and in response to the UV-C treatment were assessed by GC-MS. A total of 73 com-pounds, divided into 6 classes, were identified: alcohols (18), esters (20), aldehydes (16), hydrocarbons (3), acids (10), and others (6) (Table 1). In addition, 17 volatile compounds were detected which were present over the whole storage period, including alcohols (3), esters (2), aldehydes (7), acids (3) and others (2) (Figure 5). These results indicated that the volatile aroma components in pepino fruit were mainly alcohols, esters and aldehydes. The results of the electronic nose indicated that ethanol, oxynitride, and sulfur compounds in the pepino fruit are the most obvious types of compound types that are affected during storage and by the UV-C treatment. The results obtained in the GC-MS analysis are, thus, similar to the E-nose results. The GC-MS data indicate that storage time and UV-C treatment increased the content of alcohol compounds, whereas thiomalic acid (a sulfur compound) was only present in the control group and was not detected at 28 days in the UV-C treatment samples.

The total content of alcohols in fruits increased with storage time, and the UV-C treatment group had the highest content (Figure 5 and Table 1**)**. The major alcohol compounds present in all pepino fruit were isoamylol, 1-penten-3-ol, and 3-methylbut-3-en-1-ol, and ethe level of the first two alcohols was higher in the 1.5 kJ/m^2^ UV-C treatment group than it was in the control group as storage time progressed (Figure 5).

Esters are a major aroma component in pepino fruit. The content of esters in fruits decreased with storage time in the control group, but increased in the UV-C treatment group. A total of 7 kinds of esters (methyl myristate, ethyl 3-hydroxyhexanoate, methyl 5-chloropentanoate, ethyl 4-methylpentanoate, allyl 3-cyclohexylpropionate, nonyl acetate, and isopropyl dodecanoate) were detected in the 1.5 kJ/m^2^ UV-C treatment group of fruit at 28 d of storage but were not detected in the control group. Methyl laurate and ethyl 3-hydroxybutyrate were detected in pepino fruit from both treatment groups after storage for 28 d, although their levels were higher in the UV-C-treated fruit than the control fruit.

The content of aldehydes in pepino fruit was initially high but decreased in the fruit of both treatment groups during storage. Notably, however, the UV-C treatment inhibited the decrease. A total of seven aldehydes were detected over the course of storage: (E)-2-pentenal, pentanal, heptenal, 2-hexenal, (2E,4E)-2,4-nonadienal, hexanal, and nonanal. UV-C treatment maintained the level of (2E,4E)-2, 4-nonadienal, heptenal and (2E) -2-nonenal. Furthermore, some aldehydes, such as Octanal,1, 1-diethoxydecane, and isovaleraldehyde, were identified in UV-C-treated fruit but not in 0 d and control fruit. Other compounds, such as 1-hexanoic acid, trans-hex-2-enoic acid, 1-penten-3-one, 2-ethylfuran, thiomalic acid, etc., were also identified in pepino fruit in our study. The content of these other compounds, except thiomalic acid, were found to be higher in the UV-C treatment group than in the control group.

## 4. Discussion

Pepino fruit are prone to a sharp decrease in sensory quality, firmness, and nutrients, as well as increased rot, after they are harvested and placed in storage [9,10]. UV-C has been demonstrated to be and environmentally friendly, effective approach for maintaining the postharvest quality of harvested produce, reducing chilling injury, and increasing resistance to biotic and abiotic stresses. UV-C has been demonstrated to maintain flesh firmness, aroma, color, and nutrients, and reduce decay incidence [20,32,33,34,35]. The present study investigated the effect of UV-C treatment on the postharvest quality of pepino fruit during storage. Results indicated that a 1.5 kJ/m^2^ UV-C treatment effectively maintained sensory quality ratings and fruit firmness, reduced the rate of respiration, and inhibited ethylene production in stored pepino fruit. Similar results were reported in blueberries [36], cherry tomatoes [37], strawberries [38], and peaches [39]. Notably, our research indicated that a 2.0 kJ/m^2^ and 3.0 kJ/m^2^ UV-C treatment did not have a positive effect on the maintenance of firmness and sensory quality rankings in stored pepino fruit, whereas the 3.0 kJ/m^2^ UV-C treatment enhanced the respiration rate of pepino fruit (Figure 2). These results clearly indicate that it is necessary to determine the optimum dose of UV-C needed to preserve postharvest quality of a specific fruit or vegetable, prior to advocating its use as a postharvest management strategy. A dosage effect has also been observed in the postharvest treatment of nectarine fruit, where 3.0 kJ/m^2^ effectively managed brown rot, whereas 6.0 kJ/m^2^ UV-C treatment increased the occurrence of brown rot [40].

UV-C treatment can have a positive effect on the metabolic parameters of harvested produce, such as the respiration rate and ethylene production, but can also have a positive influence on nutritional quality [13]. Ripe pepino fruit are abundant in monosaccharides, which are the main contributors to TSS [5]. TSS is the most common indicator of fruit flavor [41]. The yellowing of green produce due to the degradation of chlorophyll is also a major postharvest problem in some fruit [35]. Our study indicated that UV-C treatment of pepino fruit can inhibit the loss of TSS and chlorophyll during storage. These findings are consistent with the results found in blueberry [42], grapefruit [34], and leafy vegetables [43]. Vitamin C and phenolics are primary nutrients in fresh fruit, which enable fruit and vegetables to have a beneficial effect on human health, but also play an important role in the non-enzymatic antioxidant system of plants, which is responsible for detoxifying reactive oxygen species (ROS) in cells [35]. Previous studies have found that UV-C treatment increases the content of antioxidants, including phenolic compounds, Vitamin C, and carotenoids in grapes, button mushrooms, and other fruits and vegetables [34,44]. The same effect was also found in our study. Namely, UV-C treatment of pepino fruit resulted in the maintenance of higher levels of vitamin C, anthocyanin, flavonoids, and total phenolics at the end of 28 d of storage, relative to untreated, control fruit.

ROS is a primary contributor to the senescence of fruit and vegetables and causes oxidative damage to proteins and lipids in plant cells. Therefore, controlling ROS production and accumulation is an important strategy for delaying the senescence of fruit and vegetables during postharvest storage [39]. MDA is a by-product of lipid peroxidation and serves as a good indicator of the degree of oxidative stress and membrane structural integrity in plants [45]. Results of our study demonstrated that the prescribed UV-C treatment (1.5 kJ/m^2^ UV-C) decreased MDA levels in pepino tissues. These results indicate that the prescribed UV-C treatment of harvested pepino may help to maintain membrane integrity and reduce ROS damage.

In addition to the non-enzymatic antioxidant system, an enzymatic antioxidant system, comprising POD, APX, CAT, etc., exists in plants and is responsible for detoxifying ROS [26]. Our study demonstrated that UV-C treatment can enhance the enzymatic antioxidant system in pepino fruit, including POD, APX, and CAT activity. The enhancement of the antioxidant system in stored pepino fruit helped to maintain ROS homeostasis and, thereby, potentially, increase disease resistance. UV-C treatment of mangoes [46], peaches [39], bananas [47], and other fruit has also been reported to increase disease resistance by enhancing the antioxidant system in fruit tissues, thereby regulating the metabolism of ROS, and maintaining ROS homeostasis in cells.

In addition to the nutritional quality of fruit, the composition and content of aroma- and flavor-related components are also important fruit quality traits [48] and are used to evaluate fruit storage durability [49]. Previous studies have shown that UV-C treatment can maintain the flavor of melon [50] and peach fruit [51]. The aroma of pepino fruit is green, fresh, sweet, and pleasant, and reminiscent of melon and mango fruit [6]. In our study, the ten sensors of the E-nose indicated that ethanol, oxynitride, and sulfur compounds in pepino fruit changed with storage time, and that UV-C contributed to the maintenance of the aroma found in fresh pepino fruit. Importantly, the results obtained with the E-nose were similar to those obtained using HS-GC-MS analysis. Alcohols, esters, and aldehydes are considered the most important aromatic compounds contributing to the perception of the aroma of fresh fruit [52]. Alcohol compounds, such as 3-methyl-2-buten-1-ol (also known as prenol), 3-methyl-3-buten-1-ol, and (Z)-6-non-1-ol, have been reported to be associated with the fresh taste of pepino and cucumber fruit [5,6,53]. The UV-C treatment, however, did not significantly increase the content of these compounds, possibly because the UV-C treatment may maintain the aroma of fruits by promoting the synthesis of other aromatic alcohols, such as hexanol, 3-methyl-2-Butanol, cis-2-Penten-1-ol, and others. Rodríguez-Burruezo et al. [54] and Shiota et al. [6] found that C6 and C9 aldehydes contribute to the green odor of fruit, and that esters, primarily nonanal, hexanol, (2E)-2-nonenal, 2-hexenal,cis-2-nonen-1-ol, and 3-methyl-3-buten-1-yl acetate, are the main contributors to the peculiar, pleasant odor of pepino fruit. All of these compounds were detected in our present study. The UV-C treatment only maintained the content of (2E)-2-nonenal and (2E)-2-nonenal in fruits, but the total content of alcohols, esters, and aldehydes in fruits was higher than it was in the control group of fruit. A total of eight alcohols, six esters, and three aldehydes were found to be unique to UV-C treated fruit. We suggest that UV-C may potentially maintain fruit flavor because UV-C promotes the accumulation of (2E)-2-nonenal, (2E)-2-nonenal, and the synthesis of other aromatic flavor compounds, thus, maintaining the total content of flavor compounds in pepino fruit. Notably, in addition to alcohols, esters, and aldehydes, the level of acid compounds in UV-C-treated fruit was higher than it was in control fruit at the end of storage. Wang et al. [55] reported that the formation of aldehydes in fruit is due to the automatic oxidation of unsaturated fatty acids. Lalel et al. [56] reported that carboxylic acids can combine with sugars to form esters, which increase the aroma of fruit and vegetables. Thus, we speculate that UV-C treatment maintains the levels of acids, aldehydes, and ester compounds in fruit, whereas at the same time, it promotes the synthesis of a large amount of alcohol and acid compounds that are converted to aldehyde compounds and ester compounds, respectively. These enhancements result in the maintenance of fruit flavor and quality during storage. Furthermore, Chen et al. [17] found that ketones and some furans represent fruit flavor enhancers. Our results also indicated that the prescribed UV-C treatment of pepino fruit maintained the level of 1-penten-3-one and 2-ethylfuran, which presumably contributed to the maintenance of the aroma of pepino fruit aroma during storage. In summary, 1.5 kJ/m^2^ UV-C treatment contributed to the maintenance of the flavor and aroma of pepino fruit during storage.

## 5. Conclusions

Results of the present study indicate that a 1.5 kJ/m^2^ dose of UV-C irradiation administered to harvested pepino fruit had a positive effect on the maintenance of sensory quality, antioxidant activity, and flavor quality during postharvest storage. The 1.5 kJ/m^2^ UV-C treatment effectively delayed fruit senescence, maintained fruit firmness, reduced the respiration rate, inhibited ethylene production, and maintained the nutritional quality of pepino fruit in storage. The prescribed UV-C treatment also reduced ROS damage by increasing the level of antioxidant compounds and antioxidant enzyme activity, and reducing MDA levels. Collectively, our results indicate that a 1.5 kJ/m^2^ UV-C treatment can be used as an effective method to maintain the quality of pepino fruit during storage.

## Figures and Tables

**Figure 1 foods-10-02964-f001:**
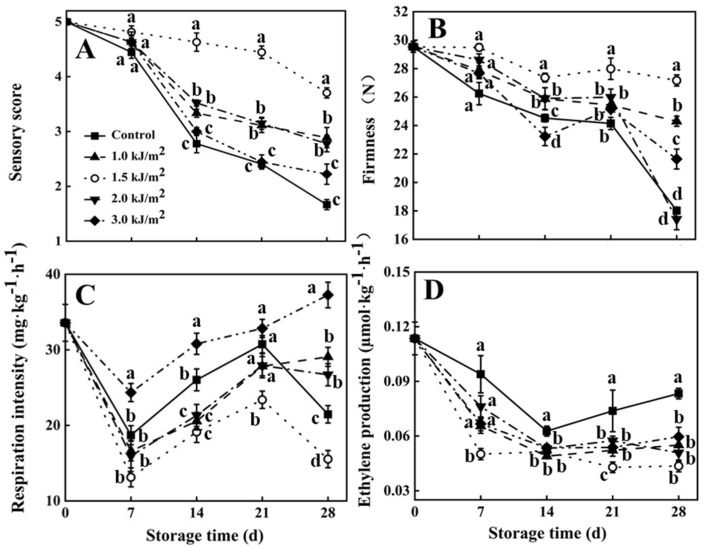
Sensory score (**A**), firmness (**B**), respiration rate (**C**), and ethylene production (**D**) of pepino fruit treated with different UV-C dose during storage. Data represent the mean ± SE (*n* = 3). Different letters indicate significant differences (*p* < 0.05) between sample groups at the time of sampling.

**Figure 2 foods-10-02964-f002:**
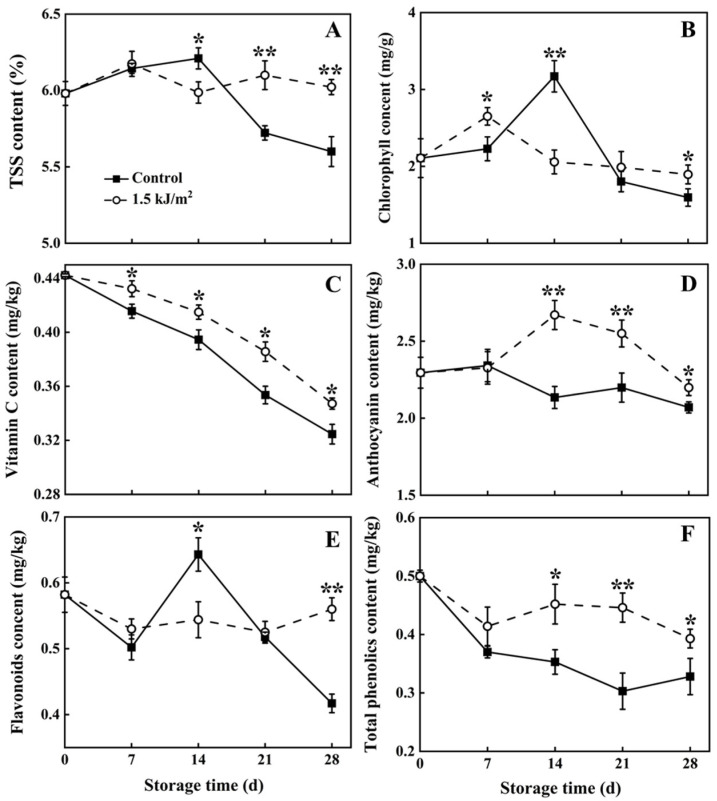
TSS (**A**), chlorophyll (**B**), vitamin C (**C**), flavonoids content (**D**), anthocyanin (**E**) and total phenolic content (**F**) of pepino fruit treated with dark (control) and UV-C during storage. Chlorophyll, vitamin C, and total phenolic content were measured in pulp tissue; flavonoid and anthocyanin content were measured in peel tissues. Asterisk (*) indicates a significant difference between the control and UV-C treatment groups at *p* < 0.05. Whereas a double asterisk (**) indicates significance at *p* < 0.01.

**Figure 3 foods-10-02964-f003:**
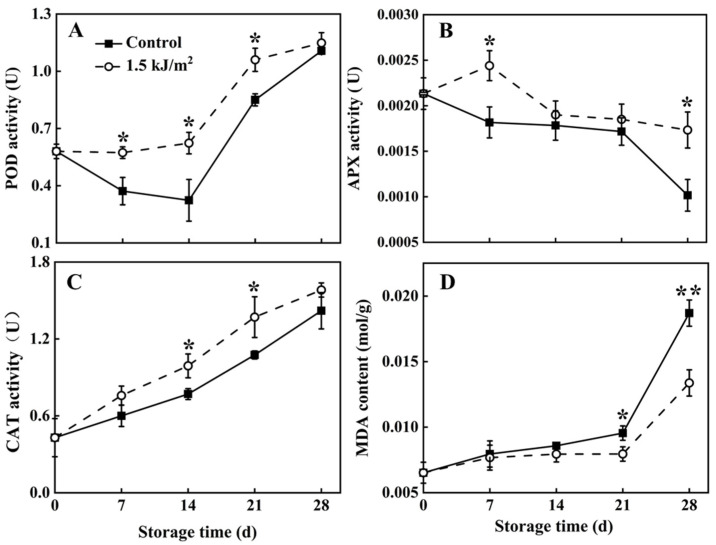
POD activity (**A**), APX activity (**B**), CAT activity (**C**) and MDA content (**D**) of pepino fruit treated with dark (control) and UV-C during storage. POD, APX and CAT activity were measured in pulp tissue; MDA content was measured in peel tissue. Asterisk (*) indicates a significant difference between the control and UV-C treatment groups at *p* < 0.05. Whereas a double asterisk (**) indicates significance at *p* < 0.01.

**Figure 4 foods-10-02964-f004:**
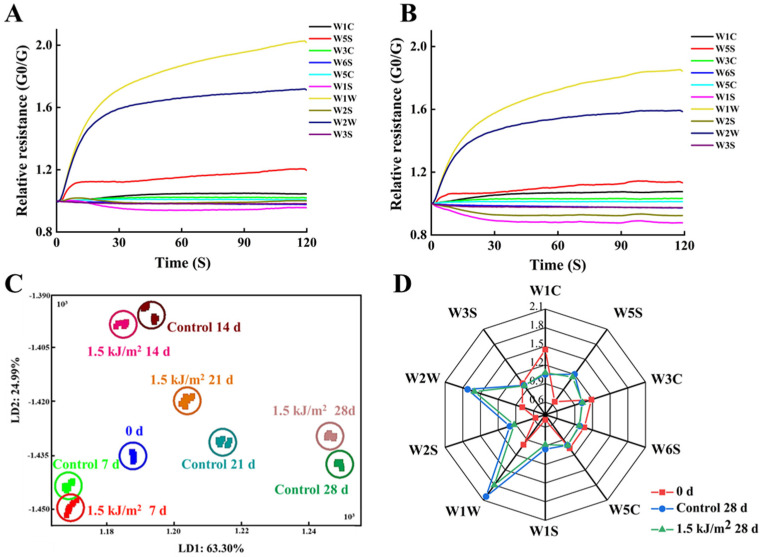
E-nose sensor intensity for volatile compounds in control pepino fruit (**A**) and 1.5 kJ/m^2^ UV-C treated pepino fruit (**B**) harvested for day 28. (**C**) Linear discriminant analysis of E-nose data for control pepino fruit and 1.5 kJ/m^2^ UV-C treated pepino fruit harvested at different time. (**D**) Radar fingerprint chart of volatile compounds in control pepino fruit and 1.5 kJ/m^2^ UV-C treated pepino fruit harvested for 28 d. Pulp tissues were used to obtain the E-nose profiles.

**Figure 5 foods-10-02964-f005:**
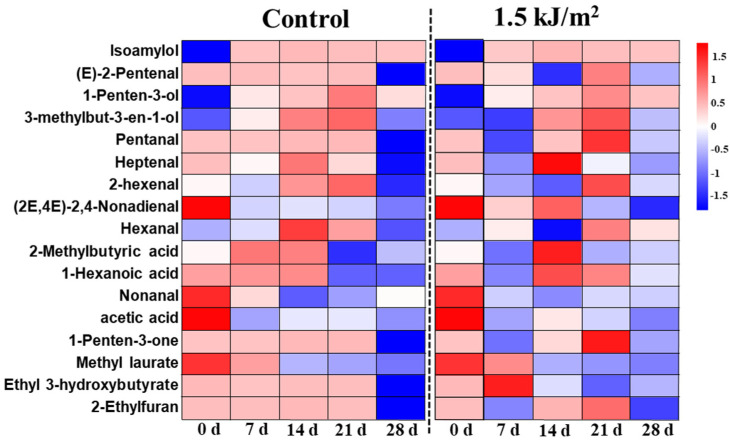
Changes in the level of flavor compounds in pulp tissues of pepino fruit treated with UV-C tand control fruit during storage. The median value of the graph is drawn after logarithmic transformation (log10). The redder the color, the higher the content. GC-MS analysis was conducted on pulp tissues.

**Table 1 foods-10-02964-t001:** Quantitative results of the level of aroma compounds in pepino fruit during storage for 0 d and 28 d.

Category	Number	Formula	Volatile Compounds	Content (mg·kg^−1^)		
				0 d	Control 28 d	1.5 kJ/m^2^ 28 d
Alcohols (18)	1	C_3_H_8_O	1-Propanol	1.4002 ± 0.02	ND	ND
	2	C_5_H_10_O	3-methylbut-3-en-1-ol	1.54 ± 0.07	1.6288 ± 0.03	1.3971 ± 0.04
	3	C_5_H_12_O	Pentanol	1.8347 ± 0.02	ND	ND
	4	C_6_H_14_O	Hexanol	2.1133 ± 0.06	2.7704 ± 0.02	2.8096 ± 0.03
	5	C_8_H_16_O	Oct-1-en-3-ol	1.0308 ± 0.01	0.9432 ± 0.01	ND
	6	C_4_H_10_S_2_	1,4-Butanedithiol	ND	0.8900 ± 0.04	ND
	7	C_5_H_10_O	1-Penten-3-ol	ND	3.8201 ± 0.13	4.2583 ± 0.09
	8	C_5_H_12_O	Isoamylol	ND	1.7775 ± 0.07	1.7782 ± 0.03
	9	C_9_H_18_O	cis-6-Nonen-1-ol	ND	0.9467 ± 0.01	ND
	10	C_5_H_12_O_2_	2-Isopropoxyethanol	ND	0.9435 ± 0.01	0.9977 ± 0.02
	11	C_4_H_10_O_2_	1,3-Butanediol	ND	ND	2.7975 ± 0.08
	12	C_5_H_12_O	3-methyl-2-butanol	ND	ND	4.0145 ± 0.54
	13	C_6_H_14_O_2_	1,5-Hexanediol	ND	ND	1.1312 ± 0.03
	14	C_9_H_20_O	3-ethylheptan-3-ol	ND	ND	0.9622 ± 0.01
	15	C_5_H_10_O	cis-2-Penten-1-ol	ND	ND	1.4607 ± 0.06
	16	C_9_H_18_O	cis-2-Nonen-1-ol	ND	ND	0.9458 ± 0.02
	17	C_6_H_14_O_6_	D-Sorbitol	ND	ND	0.977 ± 0.01
	18	C_10_H_18_O	(-)-α-Terpineol	ND	ND	0.9565 ± 0.01
	Total			7.919	13.7594	24.4471
Esters (20)	19	C_7_H_12_O_2_	3-Methyl-3-buten-1-yl acetate	1.1235 ± 0.04	1.0446 ± 0.05	ND
	20	C_10_H_20_O_2_	Methyl nonanoate	0.9666 ± 0.03	ND	ND
	21	C_11_H_22_O_2_	Methyl Caprate	1.1157 ± 0.01	ND	ND
	22	C_12_H_24_O_2_	Hexyl hexanoate	1.0269 ± 0.03	1.0348 ± 0.04	0.9600 ± 0.00
	23	C_15_H_30_O_2_	Methyl myristate	1.5104 ± 0.05	ND	1.0001± 0.02
	24	C_6_H_12_O_3_	Ethyl 3-hydroxybutyrate	1.1658 ± 0.08	0.0018 ± 0.00	1.0259± 0.03
	25	C_9_H_16_O_4_	Diethyl dimethylmalonate	1.2418 ± 0.03	ND	ND
	26	C_7_H_14_O_3_	Methyl 5-methoxypentanoate	1.1747 ± 0.03	ND	ND
	27	C_5_H_10_O_2_	Butyl formate	ND	1.0591 ± 0.02	ND
	28	C_10_H_20_O_2_	Butyl Hexanoate	ND	0.9399 ± 0.03	ND
	29	C_14_H_28_O_2_	Ethyl laurate	ND	1.5656 ± 0.03	ND
	30	C_13_H_26_O_2_	Methyl laurate	4.4772 ± 0.34	1.2296 ± 0.02	1.5939 ± 0.04
	31	C_18_H_36_O_2_	Methyl 15-methylhexadecanoate	ND	0.9945 ± 0.01	0.9530 ± 0.03
	32	C_8_H_16_O_3_	Ethyl 3-hydroxyhexanoate	ND	0.0018 ± 0.00	1.2870 ± 0.03
	33	C_6_H_11_ClO_2_	Methyl 5-chloropentanoate	ND	ND	1.1431 ± 0.05
	34	C_8_H_16_O_2_	Ethyl 4-methylpentanoate	ND	ND	1.4664 ± 0.07
	35	C_12_H_20_O_2_	Allyl 3-cyclohexylpropionate	ND	ND	0.9572 ± 0.04
	36	C_11_H_22_O_2_	Nonyl acetate	ND	ND	1.2089 ± 0.06
	37	C13H24O2	Ethyl undecylenate	ND	ND	1.2955 ± 0.05
	38	C15H30O2	Isopropyl dodecanoate	ND	ND	1.273 ± 0.03
	Total			9.3254	7.8717	14.164
Aldehydes (16)	39	C_5_H_10_O	Pentanal	1.1883 ± 0.04	0.6100 ± 0.05	1.1409 ± 0.22
	40	C_6_H_12_O	Hexanal	25.1792 ± 3.93	23.4300 ± 2.46	15.5164 ± 4.07
	41	C_5_H_8_O	(E)-2-Pentenal	1.2655 ± 0.04	ND	1.2089 ± 0.03
	42	C_6_H_10_O	2-hexenal	6.8883 ± 1.37	5.4300 ± 1.98	4.0673 ± 0.85
	43	C_9_H_14_O	(2E,4E)-2,4-Nonadienal	3.0414 ± 0.97	1.3500 ± 1.02	1.9870 ± 0.56
	44	C_7_H_12_O	Heptenal	1.2839 ± 0.32	0.3200 ± 0.02	1.9036 ± 0.36
	45	C_9_H_18_O	Nonanal	1.5303 ± 0.39	1.2569 ± 0.12	0.9751 ± 0.04
	46	C_8_H_14_O	(2E)-2-Octenal	1.4123 ± 0.06	ND	0.9724 ± 0.01
	47	C_9_H_16_O	(2E)-2-Nonenal	1.3177 ± 0.03	0.3900 ± 0.01	1.8329 ± 0.05
	48	C_9_H_14_O	(2E,6Z)-nona-2,6-dienal	1.1767 ± 0.13	ND	1.0488 ± 0.09
	49	C_7_H_14_O	Heptanal	1.0845 ± 0.04	ND	ND
	50	C_10_H_16_O	β-Cyclocitral	0.9903 ± 0.07	ND	ND
	51	C_10_H_20_O	Decanal	ND	0.2400 ± 0.01	ND
	52	C_8_H_16_O	Octanal	ND	ND	0.9943 ± 0.05
	53	C_14_H_30_O_2_	1,1-Diethoxydecane	ND	ND	1.5034 ± 0.17
	54	C_5_H_10_O	Isovaleraldehyde	ND	ND	1.8470
	Total			46.3584	33.0269	34.998
Hydrocarbons (3)	55	C_6_H_12_	cyclohexane	1.7055 ± 0.28	ND	ND
	56	C_5_H_10_	Cyclopentane	1.1283 ± 0.17	ND	ND
	57	C_10_H_16_	limonene	ND	0.9522 ± 0.01	ND
	Total			2.8338	0.9522	0
Acids (10)	58	C_2_H_4_O_2_	acetic acid	3.0367 ± 0.53	1.2403 ± 0.02	1.0919 ± 0.01
	59	C_5_H_10_O_2_	Pentanoic acid	0.9601 ± 0.01	ND	ND
	60	CH_2_O_2_	Formic Acid	0.9937 ± 0.02	ND	ND
	61	C_5_H_10_O_2_	2-Methylbutyric acid	1.1698 ± 0.30	1.1022 ± 0.05	1.0498 ± 0.02
	62	C_6_H_12_O_2_	1-Hexanoic acid	2.2315 ± 0.27	ND	3.1323 ± 0.51
	63	C_9_H_18_O_2_	Nonanoic acid	1.2055 ± 0.23	ND	ND
	64	C_9_H_16_O_2_	2-nonenoic acid	0.9751 ± 0.02	ND	ND
	65	C_4_H_6_O_4_	Succinic acid	ND	1.0831 ± 0.04	0.9740 ± 0.09
	66	C_12_H_24_O_2_	Lauric acid	ND	1.0005 ± 0.03	ND
	67	C_6_H_10_O_2_	trans-Hex-2-enoic acid	ND	1.0670 ± 0.01	1.8462 ± 0.17
	Total			10.5724	5.4931	8.0942
Others (6)	68	C_6_H_4_Cl_2_	1,3-Dichlorobenzene	1.1040 ± 0.23	0.9509 ± 0.16	ND
	69	C_6_H_8_O	2-Ethylfuran	1.1188 ± 0.15	0.0010 ± 0.00	0.9898 ± 0.02
	70	C_5_H_8_O	1-Penten-3-one	1.8548 ± 0.07	0.0020 ± 0.00	1.6495 ± 0.08
	71	C_13_H_20_O	β-ionone	0.9661 ± 0.01	ND	ND
	72	C_6_H_10_S	Diallyl sulfide	1.5546 ± 0.13	ND	ND
	73	C_4_H_6_O_4_S	Thiomalic acid	ND	2.7241 ± 0.14	ND
	Total			6.5983	3.678	2.6393

ND: Not detected

## Data Availability

Not applicable.

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
