# Peer review of "UV-C Treatment Maintains the Sensory Quality, Antioxidant Activity and Flavor of Pepino Fruit during Postharvest Storage"

_foods, 2021, doi:10.3390/foods10122964_

Round 1

Reviewer 1 Report

The Review of Manuscript Number ID:  foods-1456502 titled:

UV-C treatment maintains sensory quality, antioxidant activity and flavor of pepino fruit during postharvest storage

The manuscript has a scientific and practical dimension, contains very important issues at the problem of preserving pepino fruits, which easily lose their sensory quality, firmness and nutrients, and even rot. However, it has been proven that the treatment of 1.5 kJ / m2 dose of Ultraviolet-C UV-C limits these changes, i.e. maintains the flesh firmness, flavor, color, nutrients, has a positive effect on maintaining antioxidant activity and increases resistance to biotic and abiotic stresses and delays the aging of the fruit. incl. by reducing the respiration rate and inhibiting ethylene production. These studies may have the potential of using VC extend their shelf life of other fresh fruit or vegetables or to improve the quality of plant food raw materials. There are no publications in this section, so it is worth continuing them.

The manuscript is well prepared, it shows in-depth knowledge of the issues raised, both in terms of the research and analytical scope, as well as the discussion of the results with available literature. The research methods are relevant and well described. However, sensors scores the rating "scale was one (poor), two (average), and three 100 (good)". Why such a narrow range of the scale? How to justify such a narrow scope?

Therefore, the manuscript is prepared correctly and the issues are widely discussed. It is based on 56 references, 15 of which are from recent years (2017-2021), but older ones, from 1986, were also used. It should be emphasized that the authors very synthetically presented issues closely related to the subject of the manuscript.

Authors should only carefully read and correct the entire text of the manuscript, because punctuation errors do occur and further correct the references, i.e. in accordance with the editorial requirements, because such abbreviations as: "Herraiz et al., 2015", "Sanchez et al., 2000.", "Huyskens-Keil et al.", "Prono-Widayat et al., 2000", and write the names of journals in lowercase, e.g. agricultural research,” rather should not be used.

Author Response

Dear reviewer,

Reviewer 2 Report

In this manuscript, the authors examined the effect of UV-C treatment on the postharvest storage of pepino fruits. First, based on general quality parameters such a fruit firmness and sensory scores, the authors established 1.5 kJ/m2 of UV-C treatment as the most appropriate one to maintain fruit quality during storage. Next, the effect of 1.5 kJ/m2 on important metabolites for pepino fruit nutritional and organoleptic characteristics and on enzymatic antioxidant system was evaluated during postharvest storage. Together, the results seem to indicate that a UV-C treatment of 1.5 kJ/m2 has a positive impact on maintaining pepino fruit quality compared to untreated fruits.

The study presented here is well designed, with interesting results and conclusions. I have however a main concern about the conclusions regarding the volatile analysis (E-nose and GC-MS analyses), and I would recommend the authors to rewrite the results/discussion part about it. Indeed, based on E-nose analysis, differences between control fruits and UV-C treated fruits after 28 d of storage are not so important (Figure 4D), and the main factor responsible for sample discrimination is time of storage, not UV-C treatment (Figure 4C). 

Furthermore, regarding the affirmation in lines 449-451 of the discussion: These compounds were detected in our present study and we speculate that the prescribed UV-C treatment contributes to the accumulation of these compounds, thus, helping to maintain a green fruit aroma.

I would recommend correcting this sentence, as nonanal and 2-hexenal showed decreased levels in UV-C treated fruits compared to T0 and control fruits at 28d (see Table 1). Cis-6-nonen-1ol and 3-met-3-buten-1yl acetate were not detected in UV-C treated fruits after 28d, while detected in T0 and/or control fruits. In addition, not all the acids described in Table 1 are increased in UV-C treated fruits, so the authors should specify which ones are, and how this could lead to increased levels of aldehydes and esters. I also have a major concern about the identification of malic acid in UV-C treated fruits after 28d, compound which is undetectable in T0 and control fruits. As far as I know, this compound is generally speaking one of the main organic acid in fruits and is non-volatile. I would suggest the authors to check their data.

I also have some minor and specific comments, please see below:

Abstract

Lines 24-27: please rewrite, according to the concern about volatile analysis. In addition, a concluding sentence summarizing the study is missing.

Introduction

Line 34: please correct, to need to adapt pepino production.

Methods

Change mol L-1 to M.

Line 119: Total soluble content (TSS) was measured using a refractometer. As far as I know, the units of this measurement are in degrees Brix (º), not in percentage.

Lines 135-137: then add 1 mL AlCl3 135 and 1 mol·L-1 NaOH. This sentence is unclear, what is the concentration of AlCl3 and what volume of NaOH 1M should be added?

Lines 139-148: please rewrite, the paragraphs are not well explained, the imperative should not be used to describe methods.

Lines 205-208: please rewrite this sentence, the meaning is unclear.

Results

Lines 221-223: Please rewrite the sentence, as the meaning is unclear.

Figure 1: what is the meaning of CK?

Line 263: In figure 2C the authors refer to vitamin C content, not TSS.

Line 269: Please correct, Changes in the levels of flavonoids and anthocyanins

Lines 274-275: is this higher content in anthocyanins in the control group significant?

Figure 2:

Please correct legend of the y axis in 2D, Flavonoid content.

The units of flavonoid, anthocyanin and total phenolics contents are unclear, as in methods it is stated that flavonoid content was expressed in mg of epicatechin equivalent per kg of fruit weight (mg·kg-1), anthocyanin content was expressed in mg cyanide-3-O-glucoside equivalent per kg fruit weight (mg·kg-1), and the total phenol content in mg gallic acid equivalents per one kg fruit weight (mg·kg−1).

In addition, the letters indicating statistical differences are incomplete. As it is a comparison between control 1.5 kJ/m2 UV-C treatment, I would recommend using asterisks.

Lines 322-323: please correct the sentence.

Lines 325-327: I would say, however, that time of postharvest storage is the main factor responsible for differences in aroma profiles, based on the LDA analysis.

Figure 5: do the volatile values represent the log2-fold change compared to T0 samples? Please specify as this is unclear, which makes the interpretation of the heatmap complicated.

Author Response

Dear reviewer,

Reviewer 3 Report

Dear Editor, in the manuscript foods-1456502 authors evaluated the effect of UV-C treatment on maintaining sensory quality, antioxidant activity and flavor of pepino fruit during postharvest storage. In a previous paper of the authors (Journal of South China Agricultural University, 2021, 42(5), pp. 87-6) 1.00, 0.50 and 0.25 kJ·m−2 UV-C treatments were applied and the best results on reducing clulling injury and maintained flavor quality of pepino fruit were obtained with 1 kJ·m−2 UV-C treatment. This paper should have been cited in introduction and results section in order to highlight the new information provided in the present manuscript.

The major concerns of the present manuscript are listed below:

- Line 88: Were the pulp and peel samples of each of the 9 fruits frozen and stored separately or were they mixed to obtain a homogeneous sample of these 8 fruit?

- Line 118: How many fruit were used to measure these parameters?

- It should be clarified, along material and methods section the number of fruit that were taken for each sampling date and how many fruit were used to measure each parameter. In addition, it is important to clarify if the whole experiment was replicate. In figure legends it is stated that n=3 but it is not clear if data were the mean of determination made in 3 independent fruits o in 3 independent experiment. This is an important issue that should be clarified in the revised manuscript.

- There are also some problems with the used units. For instance, according to line 117, ethylene was expressed and µmol kg-1 h-1 but in figure 1D ethylene is expressed as µL kg-1 h-1. According to line 125, chlorophyll content was expressed as g kg-1 and it was expressed as mg g-1 in figure 2B. The same for Fig 2D, 2E and 2F. Check carefully units along the manuscript.

Other minor points are listed below:

- Lines 263-264: Remove this sentence because it has been written above.

- Line 268: Figure 2C should be cited at the end of the sentence.

- In all figure legends it should be indicated the parameters that were measured in peel or in flesh.

- Line 303: This sharp increase cannot be observed in Fig. 3D.

- Line 412: Reference number 34 is not appropriate to support this statement.

- Check reference list and write them according to the journal format.

Author Response

Dear reviewer,

Round 2

Reviewer 3 Report

The manuscript has been improved according to the reviewers' suggestions and it could be suitable for publication in its present form.

Author Response

Dear reviewer,

Thank you for your general positive comments. We appreciate your comments and suggestions and we will continue to work hard to make the article better.

Best Regards.

Yours sincerely,

Yaqi Zhao